# Wealth Status and Health Insurance Enrollment in India: An Empirical Analysis

**DOI:** 10.3390/healthcare11091343

**Published:** 2023-05-07

**Authors:** Preshit Nemdas Ambade, Joe Gerald, Tauhidur Rahman

**Affiliations:** 1Department of Population Health Sciences, Medical College of Georgia, Augusta University, Augusta, GA 30912, USA; 2Department of Community, Environment & Policy, Mel and Enid Zuckerman College of Public Health, University of Arizona, Tucson, AZ 85721, USA; 3Department of Agricultural and Resource Economics, College of Agriculture and Life Sciences, University of Arizona, Tucson, AZ 85721, USA

**Keywords:** health insurance, wealth, health policy, population health, health inequity, health research, public policy, India

## Abstract

Since 2005, health insurance (HI) coverage in India has significantly increased, largely because of the introduction of government-funded pro-poor insurance programs. As a result, the determinants of HI enrollment and their relative importance may have changed. Using National Family Health Survey (NFHS)-4 data, collected in 2015–2016, and employing a Probit regression model, we re-examine the determinants of household HI enrollment. Then, using a multinomial logistic regression model, we estimate the relative risk ratio for enrollment in different HI schemes. In comparison to the results on the determinants of HI enrollment using the NFHS data collected in 2005–2006, we find a decrease in the wealth gap in public HI enrollment. Nonetheless, disparities in enrollment remain, with some changes in those patterns. Households with low assets have lower enrollments in private and community-based health insurance (CBHI) programs. Households with a higher number of dependents have a higher likelihood of HI enrollment, especially in rural areas. In rural areas, poor Scheduled Caste and Scheduled Tribe households are more likely to be enrolled in public HI than the general Caste households. In urban areas, Muslim households have a lower likelihood of enrollment in any HI. The educational attainment of household heads is positively associated with enrollment in private HI, but it is negatively associated with enrollment in public HI. Since 2005–2006, while HI coverage has improved, disparities across social groups remain.

## 1. Introduction

India has adopted HI as a healthcare financing tool to achieve Universal Health Coverage (UHC). The growing and diverse HI sector is served by multiple players who provide a variety of HI products. The central government health insurance scheme (CGHS) and employee state insurance scheme (ESIS) cover government and private sector employees, respectively. The community-based health insurance (CBHI) programs, mediated by non-profit organizations, serve poor socioeconomic groups. The Rashtriya Swasthya Bima Yojana (RSBY), a federal program that has been renamed Pradhan Mantri Jan Aarogya Yojana (PM-JAY), and state government HI programs target poor households. India started liberalizing its HI market in 1999 for foreign investments. As a result, private HI has grown [1].

Limited government funding and the cost of HI are among the major impediments to UHC [2]. HI has high demand among poor households because it mitigates the adverse income effect of illness [3,4]. For low-income households, HI is an essential product, and they are willing to pay for it [5]. However, they cannot afford it. As a result, in response to health shocks, they often turn to risky ways to meet healthcare costs [6]. Beyond low income and affordability, lack of awareness and procedural difficulties are among the determinants of HI enrollment [7].

Using NFHS data collected in 2005–2006, Chakrabarti and Shankar [8] explore the determinants of household HI enrollment in India. They find that household asset-holding is positively associated with enrollment in HI programs. Among other determinants, they find significant roles of access to media and education. Lower-caste households with formal employment were more likely to have employer-based HI. In contrast, they were less likely to be enrolled in private HI than the upper caste households. Muslim households were less likely to have HI. They also documented a significant urban–rural gap in HI enrollment.

Since the introduction of RSBY and other state HI schemes, access to HI by poor households has improved. However, regional-level studies in India find that households from the lowest wealth quintile are less likely to be enrolled under the RSBY [9,10] partly because RSBY rollout favored the districts with fewer low-income households and lower-caste households [11]. The early implementation of RSBY faced administrative challenges such as identifying poor households [12]. As a result, the targeting of beneficiaries was imprecise [11,13,14]. However, the increase in HI coverage at the national level is largely because of RSBY and state programs [15]. However, the RSBY and other HI programs have undergone changes in recent years; for example, several states updated their below-poverty-line (BPL) list of households [12]. Further, RSBY was extended to the Mahatma Gandhi National Rural Employment Guarantee Act (MGNREGA), a federal rural employment program [16]. At the same time, states like Maharashtra managed to cover nearly 85 percent of their population under the state HI program [17]. As a result, the HI coverage among the lowest three wealth quintile households went from less than 3% to approximately 82% during 2005–2016 [18,19]. HI enrollments have also increased among Scheduled Caste (SC) and Scheduled Tribe (ST) households [14,20]. Moreover, 17 states and four Union Territories have HI programs targeting low-income households [20]. PM-JAY aims to reach nearly 10 million households [21,22].

With a significant increase in HI coverage in India since 2005–2006, there is a need for reassessing the determinants of HI enrollment utilizing the latest available data. In doing so, one can examine whether the relative roles of the determinants of HI enrollment have changed. Thus, in this paper following Chakrabarti and Shankar [8] (CS, henceforth), we re-examined the determinants of HI enrollment. We also provided the latest estimates of HI enrollments for low-income groups across different insurance products. We utilized NFHS-4 data that was collected in 2015–2016 and the Probit regression model to explore the determinants of household HI enrollment. Then, using multinomial logistic (MNL) regression, we estimated the adjusted relative risk ratios (RRR) to compare the enrollment under different HI products against non-enrollment.

Relative to 2005–2006, we find that the wealth gradient has decreased, especially for enrollment in public HI. However, wealthier households still have an advantage over poor households in private HI enrollment. Access to newspaper and television are determinants of HI enrollment. In 2015–2016, households with higher dependency and lower castes had relatively higher HI enrollment in rural areas. However, their enrollments in private HI were still low. The household head’s education was positively associated with private HI enrollment in both urban and rural areas. Muslim households were less likely to have any HI. The public HI coverage improved in rural areas, which has narrowed urban–rural disparity.

## 2. Materials and Methods

### 2.1. Data

We use de-identified data from the NFHS-4. With a nationally representative sample of households, it has an almost 98% response rate and covers various health topics [19]. The survey uses stratified two-stage sampling for the data collection where rural villages and Census Enumeration Blocks (CEB) are Primary Sampling Units (PSUs). More information on the survey’s sampling procedure is available in the NFHS-4 national report [19]. This data is available for download from the United States Agency for International Development (USAID)’s Demographic and Health Survey (DHS) Program and the International Institute for Population Sciences (IIPS) websites [23,24]. Our study uses data on 21,592 urban and 51,506 rural households across 29 states and seven union territories.

### 2.2. Description of the Variables

Since we closely follow the study by CS, our variables and model specifications are defined accordingly. We utilize information collected in households and eligible women’s (women aged between 15 and 49 years) modules of the survey. We briefly describe our outcome and explanatory variables (see Appendix A for additional details).

#### 2.2.1. Health Insurance Enrollment

Our outcome variable of interest is a binary (yes/no) variable, which takes the value of 1 if any usual member of the household is enrolled in any HI scheme at the time of the survey; otherwise, it takes a value of 0. Our sample does not include households with the response “do not know” (approximately 0.60% of the *n* = 601,509 surveyed households). For households reporting HI enrollment by any of their members (155,531), we define a group variable for HI programs considering their operational mechanism and target population. For instance, we combine ESIS, CGHS, state health insurance schemes, and RSBY as publicly funded HI schemes. The private insurance category includes privately purchased commercial plans and medical reimbursement from the employer [25], while CBHI is considered a distinct insurance choice. The rest of the HI types are grouped under the “other” category. A household not having any HI is categorized as “no-insurance” and serves as our baseline group. Thus, the group variable takes the values: “0 = no-insurance”, “1 = public health insurance”, “2 = community-based health insurance”, “3 = private health insurance”, and “4 = Others”. We exclude households who responded “more than one health insurance” from our analysis, which accounts for 4% (*n* = 6461) of the total households who reported having any kind of insurance. However, we check the robustness of our results by including these observations in the “Others” category.

#### 2.2.2. Explanatory Variables

We utilize DHS’s household wealth index variable to represent the level of household wealth. In the absence of reliable income and expenditure data, this index is useful in cross-state comparison and evaluating various public health services reaching out to the poorest of the poor [26]. Unlike income data, asset information has fewer miss-reporting chances and does not have seasonal variations [8]. Moreover, it can capture a household’s ability to pay recurring insurance premiums.

The explanatory variables include media exposure, dependency variables (no. of household members above 60 years; children 0–5 years and 6–15 years), caste status of households (along with their respective interaction dummies with the asset status), and other control variables (indicators for age, gender, religion of the household head; agriculture and non-agriculture occupation of male and female household members). Media variables capture information barriers that might hinder the uptake of HI [4,25,27]. Dependency variables capture the health risks of the non-working population with higher healthcare needs. Castes capture India’s social structure and are relevant for analysis as, historically, SCs, STs, and OBCs have worse socioeconomic and health outcomes than others. The household head’s characteristics [28] and the occupation of its members [27] are also known to be linked with the usage of the HI. More recent literature has confirmed the relevance of these variables in predicting the HI enrollment [8,10,13,29]. Among the states, we use Karnataka as the baseline, reference state. Therefore, we include dummies for the remaining states in our empirical analysis. This is also consistent with CS. We group data from Andhra Pradesh and Telangana, which were together as one state in 2005–2006. Data from union territories have been grouped into one group due to the paucity of data from each union territory. For explanatory variables as well, we exclude households with a response “don’t know”, which accounts for approximately 1 to 3 percent of the respective sample sizes. Given our large sample, it is reasonable to assume that there is no systematic bias in our estimated result. However, we perform robustness checks by including such observations. No further transformations are conducted on the included variables.

### 2.3. Empirical Strategy

Our unit of analysis is the household eligibility for HI. Since our outcome variable of the interest is a binary variable, and to maintain comparability, we also estimate the model estimated by CS. More specifically, we estimate Probit regression model. Our multivariate Probit regression model can be expressed in the equation form as follows:(1)$$Pr\left(Y=1|X\right)=\Phi (X^{\prime }\beta )$$
where Y is HI enrollment, which as defined earlier takes the value of 1 if a household is enrolled in any HI; otherwise, it takes the value of 0. Φ is the cumulative distribution function of the standard normal distribution and X represents a vector of explanatory variables.

To estimate RRRs for different HI categories against non-insurance, we estimate MNL model, which is appropriate given that we have more than two insurance products without any natural ordering, and it is also consistent with CS. For this, our equation for the base outcome (no-insurance) is:(2)$$Pr\left(y=0\right)=\frac{1}{{1+e}^{X\beta^{\left(1\right)}}+e^{X\beta^{\left(2\right)}}{+e}^{X\beta^{\left(3\right)}}{+e}^{X\beta^{\left(4\right)}}}$$

For other outcomes, it becomes
(3)$$Pr\left(y=j\right)=\frac{e^{X\beta^{\left(j\right)}}}{1{+e}^{X\beta^{\left(1\right)}}+e^{X\beta^{\left(2\right)}}{+e}^{X\beta^{\left(3\right)}}{+e}^{X\beta^{\left(4\right)}}}$$
where j=1,2,3,and 4 represents “public HI”, “CBHI”, “private HI”, and “other insurance”, respectively. Their corresponding coefficients are represented as $\beta^{\left(1\right)},\beta^{\left(2\right)},\beta^{\left(3\right)},and\;\beta^{\left(4\right)}$. Following Equation (3), the RRR for each outcome can be expressed as:$$\frac{Pr\left(y=i\right)}{Pr\left(y=0\right)}=e^{X\beta^{\left(i\right)}}$$
where *i* = 1, 2, 3, and 4 corresponds to each outcome mentioned above.

For both Probit and MNL models, standard errors are clustered at the Primary Sampling Unit (PSU) level, and a priori is set to 0.05. All the results are estimated using survey weights and strata with a single sampling unit centered at the overall mean. The analysis is performed using Stata 15.1 statistical package [30].

## 3. Results

In 2015–2016, approximately 29 percent of households had HI, with 28.8% in rural and 28.2% in urban areas [19]. Figure 1 shows the extent of HI enrollment across Indian states.

For 2015–2016, we observe a considerable variation in HI enrollment across the states. Andhra Pradesh had the highest number of insured households (74%), followed by Chhattisgarh (69%), Telangana (66%), and Tamil Nadu (64%). In Utter Pradesh, Nagaland, Jammu and Kashmir, and Manipur, less than 10% households were enrolled in any HI scheme. Figure 2 shows the distribution of various HI programs across rural and urban areas.

Enrollment in RSBY accounts for the highest share of HI enrollments. The reach of public HI schemes is higher in rural areas, while ESIS, CGHS, and private HI have higher enrollment in urban areas. The enrollment in CBHI schemes in 2005–2006 was 12.07 and 2.73 percent in urban and rural areas, respectively [8]. Table 1 provides a list HI programs and their eligibility criteria.

Table 2 shows the distribution of HI enrollment by potential explanatory variables. Except for newspaper, household head’s education, and regional dummy, all other variables have a statistically significant association with the HI enrollment. All explanatory variables included in the analysis share statistically significant associations with HI choices. We do not exclude any variables from further analyses due to their theoretical importance and to maintain comparability with CS.

In Table 3, we compare our findings with that by CS. Appendix B provides a detailed comparison.

### 3.1. Determinants of HI Enrollment

The marginal effects of wealth status, media, age profile, caste, and other covariates estimated using the Probit model for rural, urban, and combined samples are presented in Table 4.

#### 3.1.1. Role of Household Wealth

Unlike CS’s results, we do not find a significant impact of a household’s wealth status on its current HI enrollment in rural and urban areas. Compared to 2005–2006, the relative advantage of the wealthier households has decreased across the rural and urban areas. However, after controlling for residence, the highest asset group had a higher probability of having HI than the low wealth group (4 percent higher probability).

#### 3.1.2. Role of Media

Our result is consistent with CS. Among all media variables, CS reports the smallest marginal effects for the radio variable. Similarly, our results are small but insignificant for the variable. The effects of newspaper and television are significant in urban and rural areas, respectively. An urban household with any adult woman reading a newspaper at least once a week had a 2.2 percentage point higher probability of having a member enrolled in HI. Similarly, a rural household with any adult woman watching television at least once a week had a 2.8 percentage point higher probability of having any member enrolled in a HI scheme. Like CS, to isolate the effect of media variables from education and wealth status, we re-estimated our Probit model by including predicted residuals of each of the media variables (results available upon request). The results for media variables remained the same as in our primary model. Thus, the effect of access to media variables persisted after controlling for education and wealth status. Therefore, as CS has noted, insurance providers’ better advertising and marketing strategies would help reach yet-to-be-insured households, providing them more access to the information on the offered health insurance products.

#### 3.1.3. Role of Dependency Variables

Contrary to CS, we find that in 2015–2016, both in urban and rural areas, households with a higher number of older adults had a higher probability of enrollment in any HI. This suggests that in the era in which RSBY and state-funded programs have been introduced, such households might have enrolled in these programs anticipating greater healthcare need. However, similar to CS, the interaction dummy for high assets and no. of the elderly is insignificant, suggesting no joint effect of high wealth and the high number of older adults on the enrollment.

We find small marginal effects of the number of children in the household. In the urban areas, the probability of HI decreases with the increasing number of children (significant for the 0–5 years group at 2.7 percentage points). In the rural area, we find a small marginal but positive effect (0.5 percent) of the variable representing children aged 6–15 years old. This is consistent with the literature [28,31,32,33], which found a positive association between age and demand for HI.

#### 3.1.4. Role of Caste

Consistent with CS, in urban areas, we find SC, ST, and OBC (relative to the base category upper caste) are statistically insignificant determinants of HI enrollment. Neither of the lower-caste households with higher assets have significantly different enrollments than the low-asset households from the same castes. However, in rural areas, SC and ST households have a higher likelihood of HI enrollment (5.5 percent and 3.4 percent, respectively). This result indicates that HI enrollments of SC and ST households with low-income have improved in the rural areas. In contrast to CS, the caste–wealth interaction dummies are statistically insignificant. In the full sample (rural and urban data combined), the SC and OBC households with higher assets have lower likelihoods of HI enrollments.

#### 3.1.5. Role of Other Control Variables

We find a significant but small positive marginal effect of the household head’s education on HI enrollment in the urban area. Similar to CS, the likelihood of HI enrollments has not improved for the urban Muslim households compared to other minority religious groups. Hindu households in rural areas have a higher likelihood of HI than minority religious groups’ households. The occupational status of male members of the household has no significant effect on HI enrollment. However, the occupational status of female members has significant effects (except for non-agriculture occupation in the urban sample).

CS documented negative coefficients for state dummies, suggesting households living in other states in comparison to Karnataka had lower likelihoods of HI enrollment. In contrast, we find positive coefficients for states such as Andhra Pradesh, Arunachal Pradesh, Chhattisgarh, Kerala, Mizoram, and Tamil Nadu in urban areas (results not shown). In addition, in contrast to CS, we find a significantly lower likelihood of HI enrollment for urban households (4.1 percent), accounting for other factors. The programs like RSBY and state-funded health insurance programs focus on poor households predominantly living in rural areas.

### 3.2. Determinants of HI Enrollment by Schemes

The results from the estimation of MNL are presented in Table 5. As highlighted earlier, with RSBY and state-funded HI programs, public HI became the major category of HI schemes in 2015–2016. Therefore, we present the results for public HI schemes along with CBHI and private HI schemes. We examine the differential impact of household wealth, access to media, demographics, and location on alternative HI enrollment. Due to data limitations, we combine CBHI with “Other” HI in the analysis of the urban area.

#### 3.2.1. Role of Household Wealth

CS documented that households (both low and high caste) with higher asset holdings were more likely to be enrolled in public, CBHI, and private HI schemes, implying enrollment gaps in such schemes between poor and rich households. In contrast, we find this result only for private HI scheme.

#### 3.2.2. Role of Media

Households with access to television at least once a week have a higher likelihood of having public HI in rural areas (RRR 1.2) and private HI in urban areas (RRR 3.39) compared to households without any HI. Reading newspapers has a significant effect on enrollment in private HI vis-à-vis no insurance in the urban areas.

#### 3.2.3. Role of Dependency Variables

Similar to CS, we do not find a significant role of high wealth and high number of older adults on any type of HI enrollment. Additionally, the negative association between the presence of children 0 to 5 years is the same as what CS reported. However, our results for other dependency variables are in contrast to CS. In rural areas, households with members older than 60 years and 6–15 years old have a higher likelihood of enrollment in public HI (RRR 1.08 and 1.04, respectively).

#### 3.2.4. Role of Caste

In urban areas, both low- and high-caste households have a comparable likelihood of being enrolled in public HI. Further, the interaction effect between assets and low-caste variables are insignificant, which is contrary to CS. However, SC and OBC poor households have a higher likelihood of being enrolled in CBHI or private HI when compared with poor upper-caste households. The high-asset SC(OBC) households have a higher probability of enrolling in CBHI and private HI when compared with general-caste and low-asset SC(OBC) households. Contrary to CS’s results, the enrollments for rural SC and ST households are higher in public HI (respective RRRs 1.41 and 1.25). Additionally, medium-asset ST (OBC) households are more likely to be enrolled in CBHI than low-asset ST (OBC) households. Overall, we report contrasting results with CS for the interaction dummies of caste and wealth status.

#### 3.2.5. Role of Other Control Variables

Consistent with CS, we find that Muslim households are less likely to be enrolled in any HI compared to non-Muslim households (significant for CBHI enrollment in urban areas). Rural Hindu households are more likely to been enrolled under public HI than the minority religious groups (RRR 1.32). Consistent with CS, we find that higher educational attainment of the household head is positively associated with the likelihood of private HI enrollment. Similarly, we find men’s non-agriculture occupation is positively associated with private HI enrollment in urban areas. In the rural areas, except for public HI, member occupations do not significantly affect household enrollment in any HI category. More specifically, women’s occupation is positively associated with public HI. The region dummy, which checks urban–rural differences, indicates a higher probability of public HI in rural households. Conversely, urban households have a higher likelihood of private HI enrollment.

## 4. Discussion

Our reassessment of the determinants of household HI enrollments show that the roles of household wealth, dependent members, and caste have changed. Poor and lower-caste households are more likely to be enrolled, particularly in public HI programs. Access to media and household head characteristics remain important predictors of HI enrollment. Muslim households continue to have a lower likelihood of enrollment in HI programs. The enrollment momentum shifted from the urban areas in 2005–2006 to the rural areas in 2015–2016. Our results suggest that the increase in HI enrollments can be largely attributed to the introduction of pro-poor public HI programs since 2005.

In contrast to CS, we find that the likelihoods of public HI enrollments of wealthier and poor households are approximately the same. This is likely because the enrollment criteria for RSBY and state-funded HI schemes differ. In some states, the HI schemes allow non-poor households; for example, Andhra Pradesh (including Telangana) and Tamil Nadu use their own list of poor households and cover 80 and 50 percent of their population, respectively [15]. However, high-asset households have a higher likelihood of enrollment in CBHI and private HI.

We find that access to the newspaper and television are significant predictors of HI enrollment. Like CS, households with access to newspapers and television have a higher likelihood of enrollment in private HI. However, in rural areas, households with access to television have a higher likelihood of enrollment in public HI. Thus, CS’s finding that access to information influences voluntary HI choices is applicable in both rural and urban areas. We also find that the households with a higher number of dependents have lower likelihoods of enrollment in CBHI and private HI, a finding that is consistent with CS.

In contrast to CS, we find that SC/ST/OBC households with high-asset holding do not have a higher likelihood of enrollment in public HI. However, in rural areas, these households have a lower likelihood of enrollment in public HI. The interaction terms between high assets and low castes are statistically insignificant, which suggests that as far as the likelihood of HI enrollment is concerned, there is no meaningful difference between low-caste households with low and high assets. In urban areas, poor SC and OBC households have a higher likelihood of enrollment in CBHI compared to non-poor SC and OBC households. For private HI, poor lower-caste households, specifically SC (urban) and OBC (urban and rural), have a higher likelihood of enrollment than the poor general-caste households. The enrollment probabilities are higher for poor SC and OBC households. However, on average, high assets are positively associated with enrollment in private HI.

In urban areas, we find that household head’s educational attainment is positively associated with enrollment in any HI enrolment, a finding that is in contrast with CS. For private HI enrollment, our finding is consistent with CS and related studies [31,32,33,34,35] as the probability of enrollment increases with increasing years of education. Our education result is contradictory to CS’s findings for public HI and CBHI, as fewer years of education are linked with higher odds of enrollment in these programs. The recent literature from India [21] confirms our findings suggesting the changed relationship of this variable. Our results are consistent with the international literature [28]. We find a significantly lower probability of CBHI for urban Muslim households. For the rest of the categories, religious minority households’ likelihood of enrollment is statistically indifferent to that of Hindu and Muslim households. India’s Muslims have worse education and employment indicators than the other religious groups [36,37]. Despite significant gains in HI enrollments in the last two decades, these programs have yet to sufficiently cover Muslims, who have a greater need for HI. We find that rural households belonging to religious minorities are less likely to have public HI. PM-JAY and state programs need to cover a lot of ground to reach out to these communities. Urban households are more likely to have private HI, while rural households have a higher likelihood of public HI. This result is not surprising, as the private HI providers focus more on urban areas, whereas public HI schemes have a greater reach to households in rural areas.

In the post-RSBY period, we find that the associations between enrollments in HI schemes and some of their explanatory variables have changed, and wealth-based, ethnic, religious, occupational, and geographic disparities in enrollments still exist despite gains in the past 10–15 years. Rising morbidity and mortality, low public health expenditure, out-of-pocket expenditure, and limited coverage of then-existing insurance programs made policymakers think of comprehensive insurance programs [38]. Post 2008, public health insurance coverage increased in India because states like Tamil Nadu, Andhra Pradesh, Karnataka [39], and Maharashtra [17] implemented their own health insurance programs with more generous packages and enrollment criteria. Further, Kerala and Chhattisgarh covered their non-poor households under state-level programs [15,40].

However, increased enrollments do not necessarily translate into improved outcomes for the HI enrollees. RSBY, a major driver of increase, has been studied extensively, and evidence shows that there is some impact on health service utilization but no significant decline in out-of-pocket health expenditure [13,41,42,43,44,45]. We also detected increased institutional deliveries for the poor, viz non-poor in the post-RSBY period compared to the pre-RSBY period [46]. However, the benefits coverage was limited under the program and was mostly focused on secondary or tertiary healthcare requirements. Despite having several HI programs in the country, a study shows that buying medicine is the most important item in health-related expenditure [47]. Outpatient and medicine costs are mostly paid out-of-pocket by the patients. The revamped PM-JAY covers 3 days of pre-hospitalization and 15 days of post-hospitalization expenses on medicines and diagnostics [48]. Still, PM-JAY beneficiaries have out-of-pocket expenditure [49]. A study suggests that the program is ineffective in reducing catastrophic health expenditure [50].

Rising health inequities in India is another concern that hinders the progress toward UHC. A recent study finds that India is behind on 19 out of 33 Sustainable Development Goal (SDG) targets [51]. Seventy-five percent of districts in India are well behind the target on these indicators. Public health subsidy for treating chronic diseases in hospitals is largely utilized by the rich [52]. The life expectancy gap between the poorest and richest households is 7.6 years [53]. Moreover, geographic, ethnic, religious, and gendered health disparities are rising [54]. Caste is an important dimension to health disparities in India. Lower caste households have the worst nutritional status [55,56], life expectancy [57], infant mortality [58], and other important healthcare indicators.

Therefore, to address these inequities, public health infrastructure needs to be improved. Despite a significant gain over the past decade, the neglect of public institutions and diverting of resources to private care via health insurance programs have adversely affected the public health infrastructure. Studies suggest that such programs encourage profiteering behavior by the private health sector [13,43]. Strengthening public health infrastructure is a potentially more economical and effective option [13,43,59]. Moreover, health disparity in India ought to be examined from a socioecological framework [60] because enrollment in HI does not necessarily translate into program acceptability [61].

The coverage of CBHI declined between 2005 and 2006 and 2015 and 2016. We exclude households who reported more than one HI and “don’t know” from our main analysis. However, we perform robustness checks by re-estimating MNL results after including households with “more than one HI” in the Other HI category and estimation of Probit and MNL models by including households with “don’t know”. We find that the results do not differ qualitatively from our main analysis (see Appendix C, Appendix D and Appendix E).

Our analysis has a few limitations. To maintain the comparability with CS, we do not distinguish the RSBY and state-funded programs from the employee-targeted CBHI and ESIS within the public HI programs. Additionally, we do not analyze the inter-state variability in HI enrollment due to the desire to compare our findings with that of CS. States play an important role in the implementation of public HI programs. In recent years, some states have expanded eligibility criteria, covering almost their entire populations (see Table 1). Moreover, states are implementing PM-JAY by adopting either trust, insurance, or mixed modes of program implementation, causing variations in their program administration [48], which may have affected HI enrollments. For future research, given the recent changes in the HI sector, analyzing HI enrollment by states, ethnicity, and income will be insightful.

## 5. Conclusions

India introduced RSBY in 2008 to provide health insurance to poor households. In addition, similar state-level health insurance programs were adopted by various states. Therefore, it is reasonable to believe that the last 15 years have been more favorable to the poor as far as access to health insurance is concerned, which is also reflected in improved enrollments of poor households in different health insurance schemes. In a comprehensive study, CS explored the determinants of HI enrollment in India using data from the NFHS that was collected in 2005–206. In light of the introduction of RSBY and state-level HI schemes, it is expected that the relative roles and significance of the determinants of household enrollment in HI may have changed.

In this paper, following CS and using NFHS data that was collected in 2015–2016, we re-examine the determinants of household HI enrollment. In contrast to CS, we find that households with high assets are as likely to be enrolled in any HI as the households with low assets. Lower-caste households, especially in rural areas, have a higher likelihood of HI enrollment. Households with a higher number of dependents (i.e., elderly and children) are more likely to be enrolled in any HI. In addition, urban households are less likely to be enrolled in HI compared to rural households. On the other hand, consistent with CS, Muslim households are less likely to be enrolled in any HI compared to non-Muslim households. The educational attainment and age of the household head and women’s occupations are positively associated with enrollment in HI. Regarding enrollment in different HI programs, contrary to CS, we find that households with higher assets are as likely to be enrolled in public HI as households with low assets. Households with a higher number of dependents have a higher likelihood of enrollment in public HI. The coverage momentum has shifted to the rural areas in 2015–2016 from the urban areas in 2005–2006. While there has been a significant gain in HI enrollments, disparities across socioeconomic groups remain.

## Figures and Tables

**Figure 1 healthcare-11-01343-f001:**
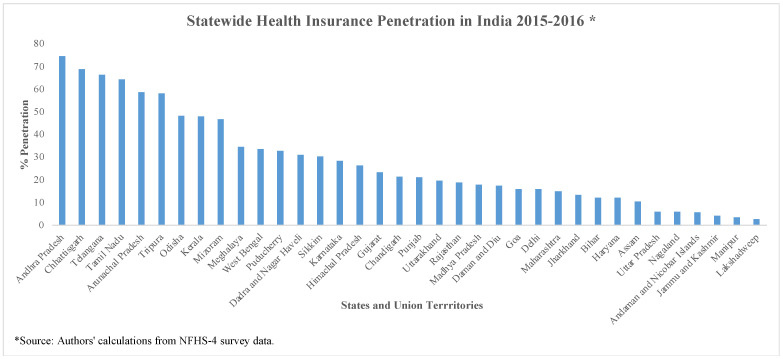
Statewide health insurance enrollment in India 2015–2016.

**Figure 2 healthcare-11-01343-f002:**
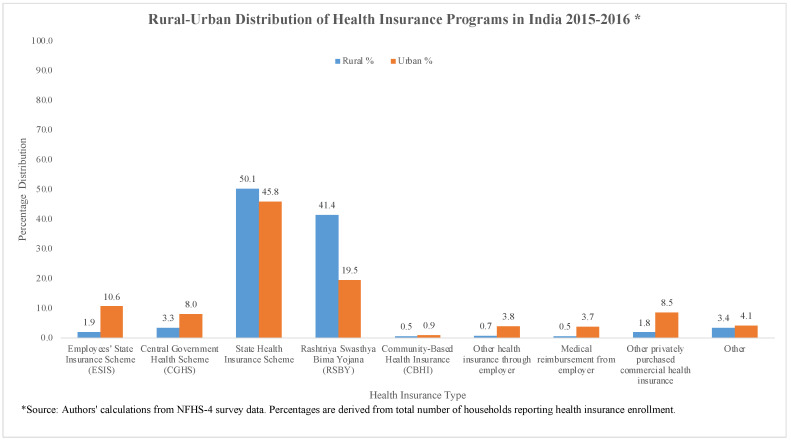
Rural–urban distribution of health insurance programs in India 2015–2016.

**Table 1 healthcare-11-01343-t001:** Central and state health insurance programs available in India.

Inception Year	Central or State	Scheme	Eligibility Criteria
1948	Central Govt.	Employee State Insurance Scheme (ESIS)	Blue-collar workers
1954	Central Govt.	Central Government Health Scheme (CGHS)	Govt. employees
2008	Central Govt.	Rashitriya Swasthya Bima Yojana (RSBY) renamed/revamped Pradhan Mantri Jan Arogya Yojana (PM-JAY) in 2018	Below Poverty Level (BPL) households working in the unorganized sector. In PM-JAY, they are identified by inclusion, deprivation, and occupational criteria of the Socio Economic Caste Census 2011 (SECC 2011).
2016	Assam	Atal Amrit Abhiyan/PM-JAY (in 2018)	Similar to PM-JAY
2015	Andaman and Nicobar Island	Andaman and Nicobar Island Scheme for Health Insurance (merged with PM-JAY in 2019)	Similar to PM-JAY
2007	Andhra Pradesh	Arogyashree Scheme (YSR Arogyasri Scheme after 2017)	State residents with an annual income below INR 500,000. Households with white ration card issued under National Food Security Act 2013.
2014	Arunachal Pradesh	The Arunachal Pradesh Chief Minister’s Universal Health Insurance Scheme (merged with PM-JAY in 2018)	Similar to PM-JAY
2013	Chhattisgarh	Mukhya Mantri Swasthya Bima Yojana (merged with PM-JAY and other programs in 2019)	Similar to PM-JAY
2013	Dadra and Nagar Haveli, Daman and Diu	Sanjeevani Swasthya Bima Yojana	Family listed in state BPL list
2016	Goa	Deen Dayal Swasthya Seva Yojana	All households
2012	Gujarat	Mukhya Mantri Amrutam Yojana	Household listed in state BPL list
2016	Himachal Pradesh	Mukhya Mantri State Health Care Scheme (merged with PM-JAY under the name Mukhya Mantri Himachal Health Care Scheme (HIMCARE) since 2019).	Similar to PM-JAY
2017	Jharkhand	Mukhya Mantri Swasthya Bima Yojana (merged with PM-JAY in 2018)	Similar to PM-JAY
2018	Karnataka	Ayushman Bharat-Aarogya Karnataka	Eligible households defined under National Food Security Act 2013 and beneficiaries for all ongoing schemes (Yashaswi Health Insurance Scheme 2003 and Vajpayee Arogyashree Scheme 2009). For non-beneficiaries, “co-payment” system is available.
2008	Kerala	Comprehensive Health Insurance Scheme (CHIS) and CHIS Plus (merged with PM-JAY in 2020 to form Karunya Arogya Suraksha Padhathi (KASP))	Similar to PM-JAY
2012	Maharashtra	Rajiv Gandhi Jeevandayee Aarogya Yojana, renamed Mahatma Jyotiba Phule Jan Aarogya Yojana in 2017	Eligible households defined under National Food Security Act 2013 having yellow and white ration cards.
2012	Meghalaya	Megha Health Insurance Scheme (merged with PM-JAY in 2019)	All residents except govt. employees
2013	Odisha	Biju Swasthya Kalyan Yojana	Eligible households defined under National Food Security Act 2013
2016	Puducherry	Puducherry Medical Relief Society Scheme (merged with PM-JAY in 2019)/state scheme	All residents except govt. employees
2015	Punjab	Bhagat Puran Singh Health Insurance Scheme (implemented PM-JAY in 2019)	Similar to PM-JAY
2015	Rajasthan	Bhamashah Health Insurance Scheme/Mukhya Mantri Chiranjeevi Swasthya Bima Yojana	All residents (no premium for socioeconomically weaker families identified under Socio Economic Caste Census 2011 (SECC 2011)).
2012	Tamil Nadu	Chief Minister Comprehensive Health Insurance Scheme	Family annual income below INR 120,000
2007	Telangana	Arogyashree Scheme	BPL families identified in state list
200	Tripura	RSBY/PM-JAY	Similar to PM-JAY
2016	Uttarakhand	Mukhya Mantri Swasthya Bima Yojana	All residents except govt. employees and pensioners
2017	West Bengal	Swasthya Sathi	Similar to PM-JAY

Note: The list is not exhaustive, and information on the schemes is collected from the respective scheme’s website.

**Table 2 healthcare-11-01343-t002:** Distribution of health insurance enrollment by potential explanatory variables.

A. Health Insurance							
			Total, N = 598,252	Has No HI, N = 425,778		Has HI, N = 172,474	*p*-Value
Household Asset Variables (n = 598,252)							<0.001
High Asset			31.424	30.676		33.269	
Medium Asset			23.801	22.622		26.713	
Low Asset			44.775	46.702		40.018	
Media Exposure Variables (n = 482,158)							
Newspaper			44.247	42.706		48.046	<0.001
Radio			18.141	18.033		18.408	0.14
Television			78.155	74.688		86.699	<0.001
Dependency Variables (n = 598,252)							
Prop. of # above 60			0.392 (0.647)	0.390 (0.649)		0.397 (0.642)	<0.001
Prop. of # between 0 to 5			0.512 (0.831)	0.551 (0.860)		0.416 (0.746)	<0.001
Prop. of # between 6 to 15			0.925 (1.156)	0.974 (1.195)		0.804 (1.045)	<0.001
Caste (n = 571,188)							
SC			21.592	20.962		23.126	<0.001
ST			9.642	9.388		10.259	<0.001
OBC			44.241	43.305		46.518	<0.001
Control Variables							
Hindu (n = 598,252)			81.383	80.020		84.746	<0.001
Muslim (n = 598,252)			12.568	14.105		8.772	<0.001
Sex of Household Head (n = 598,252)							0.003
Female			14.636	14.777		14.290	
Male			85.364	85.223		85.710	
HH head’s age (n = 598,168)			48.415 (14.024)	48.017 (14.291)		49.398 (13.292)	<0.001
HH head’s education (n = 595,856)			6.044 (5.208)	6.042 (5.172)		6.048 (5.294)	0.5
Male: Agriculture (n = 78,207)			34.268	33.335		36.387	<0.001
Male: Non-Agriculture (n = 78,207)			65.562	66.273		63.949	0.001
Female: Agriculture (n = 82,550)			18.203	16.848		21.313	<0.001
Female: Non-Agriculture (n = 82,550)			18.511	16.954		22.082	<0.001
Region (n = 598,252)							0.054
Rural			65.163	64.951		65.686	
Urban			34.837	35.049		34.314	
**B. Health Insurance Products**							
	**Total, N = 591,378**	**Has no HI, N = 425,778**	**Has Public HI, N = 148,369**	**Has CBHI, N = 809**	**Has Private HI, N = 10,823**	**Has Other HI, N = 5599**	***p*-Value**
Household Asset Variables (n = 591,378)							<0.001
High Asset	31.256	30.676	28.511	49.026	83.453	44.596	
Low Asset	44.895	46.702	43.142	25.893	5.764	32.284	
Medium Asset	23.850	22.622	28.347	25.081	10.782	23.120	
Media Exposure Variables (n = 476,354)							
Newspaper	44.092	42.706	44.989	60.154	79.834	56.352	<0.001
Radio	18.072	18.033	17.201	24.765	29.021	22.643	<0.001
Television	78.016	74.688	85.894	88.524	95.996	86.723	<0.001
Dependency Variables (n = 591,378)							
Prop. of # above 60	0.392 (0.647)	0.390 (0.649)	0.390 (0.634)	0.406 (0.663)	0.448 (0.696)	0.496 (0.715)	<0.001
Prop. of # between 0 and 5	0.513 (0.832)	0.551 (0.860)	0.419 (0.749)	0.452 (0.764)	0.362 (0.672)	0.448 (0.786)	<0.001
Prop. of # between 6 and 15	0.927 (1.158)	0.974 (1.195)	0.815 (1.053)	0.792 (0.997)	0.655 (0.913)	0.829 (1.074)	<0.001
Caste (n = 564,457)							
SC	21.628	20.962	24.453	16.541	10.737	18.083	<0.001
ST	9.589	9.388	10.669	6.224	2.972	9.077	<0.001
OBC	44.178	43.305	47.368	45.225	36.136	40.158	<0.001
Control Variables							
Hindu (n = 591,378)	81.295	80.020	84.507	83.381	84.298	86.979	<0.001
Muslim (n = 591,378)	12.654	14.105	9.189	9.281	6.632	6.236	<0.001
HH Head’s Sex (n = 591,378)							<0.001
Female	14.673	14.777	14.803	12.565	9.764	13.082	
Male	85.327	85.223	85.197	87.435	90.236	86.918	
HH head’s age (n = 591,3778)	48.403 (14.037)	48.017 (14.291)	49.286 (13.296)	49.731 (13.175)	50.461 (13.395)	50.183 (13.399)	<0.001
HH head’s education (n = 589,002)	6.027 (5.200)	6.042 (5.172)	5.535 (5.068)	8.002 (5.693)	11.302 (4.866)	7.426 (5.310)	<0.001
Male: Agriculture (n = 77,166)	34.195	33.335	38.136	39.225	12.287	34.219	<0.001
Male: Non-Agriculture (n = 77,166)	65.629	66.273	62.169	60.702	88.122	67.039	<0.001
Female: Agriculture (n = 81,464)	18.181	16.848	22.713	9.075	5.030	21.959	<0.001
Female: Non-Agriculture (n = 81,464)	18.420	16.954	21.733	19.247	25.613	19.598	<0.001
Region (n = 591,378)							<0.001
Rural	65.234	64.951	69.206	52.477	25.144	60.862	
Urban	34.766	35.049	30.794	47.523	74.856	39.138	

Note: Column percentages are shown. # in variable names is used as a symbol for the word “number”.

**Table 3 healthcare-11-01343-t003:** Comparative summary: relative role and significance of potential explanatory variables.

	Health Insurance Enrollment		Health Insurance Choices	
	Chakravarti and Shankar [15]	Our Study	Chakravarti and Shankar [15]	Our Study
Household Asset Variables				
Medium Asset	Positive effect	No effect	Positive effect on public and private HI	Positive effect on private HI
High Asset	Positive effect	Positive effect (overall)	Positive effect on public and private HI	Positive effect on private HI
Media Exposure Variables				
Newspaper	Positive effect	Positive effect (urban and overall)	Positive effect on CBHI (urban) and private HI	Positive effect on private HI (urban and overall)
Radio	Positive effect (urban and overall)	No effect	Positive effect on private HI (urban and overall sample)	Negative effect on CBHI (rural)
Television	Positive effect	No change	Positive effect on all types of HI	Positive effect on public HI (rural and overall) and private HI (urban)
Dependency Variables				
High Asset * # above 60	No effect	No change	No effect	No change
# above 60	No effect	Positive effect	Positive effect on private HI (overall)	Positive effect on public HI (rural and overall)
# between 0 and 5	No effect	Negative effect (urban and overall)	Negative effect on public HI (urban)	No change
# between 6 and 15	No effect	Positive effect (rural)	No effect	Positive effect on public HI (rural and overall)
Caste				
SC	No effect	Positive effect (rural and overall)	No effect	Positive effect on public HI (rural and overall), CBHI (urban), and private HI (urban)
ST	No effect	Positive effect (rural and overall)	No effect	Positive effect on public HI (rural)
OBC	No effect	No change	Negative effect on public HI (urban)	Positive effect on CBHI (urban) and private HI.
SC * Medium Asset	Positive effect (rural and overall)	No effect	Positive effect on public HI (overall)	Negative effect on CBHI and private HI (urban)
SC * High Asset	Positive effect (rural and overall)	Negative effect (overall)	Positive effect on public HI (rural and overall), Negative on CBHI (rural)	Negative effect on CBHI and private HI (urban)
ST * Medium Asset	No effect	No change	No effect	Positive effect on CBHI (rural and overall)
ST * High Asset	Negative effect (overall)	No change	Negative effect on private HI (overall)	Negative effect on CBHI (rural) and private HI (urban and overall)
OBC * Medium Asset	No effect	No change	Negative effect on private HI (urban)	Negative effect on CBHI and private HI (urban) and positive effect on CBHI (rural)
OBC * High Asset	No effect	Negative effect (overall)	Negative effect on private HI (urban and overall)	Negative effect on public HI (urban) and private HI (urban and overall)
Control Variables				
Hindu	Positive effect	Positive effect (rural)	Positive effect on public HI and CBHI (overall)	Positive effect on public HI (rural)
Muslim	Negative effect (urban and overall)	No change	Negative effect on private HI (urban and overall)	Negative effect on CBHI (urban)
Female-headed HH	No effect	No change	No effect	No change
HH head’s age	Positive effect	No change	Positive effect on public and private HI	Positive effect on public HI and CBHI (rural and overall)
HH head’s education	Positive effect	Positive effect (urban)	Positive effect	Negative effect on public HI (overall), positive effect private HI
Agriculture: Male	Positive effect (overall)	No effect	Positive effect on CBHI (overall)	No effect
Non-Agriculture: Male	Positive effect (urban and overall)	No effect	Positive effect on public and private HI (urban and overall)	Positive effect on private HI (urban and overall)
Agriculture: Female	Positive effect (rural and overall)	No change	Positive effect on private HI (overall)	Positive effect on public HI (rural and overall)
Non-Agriculture: Female	Positive effect (rural)	Positive effect	Positive effect on CBHI (rural and overall)	Positive effect on public HI (rural and overall)
Region	Positive effect	Negative effect	Positive effect on public and private HI	Negative effect on public HI and positive effect on private HI

Note: Significant results are noted for samples as shown in the bracket. Mention of no sample indicates results are significant for all three samples (rural, urban, and overall). * in variable names indicates interaction, and # is used as a symbol for the word “number”.

**Table 4 healthcare-11-01343-t004:** Marginal effects of variables affecting health insurance coverage in 2015–2016 estimated using Probit model for rural, urban, and combined sample.

	Urban		Rural		Combined	
	Marginal Effects	*p*-Value ^†^	Marginal Effects	*p*-Value ^†^	Marginal Effects	*p*-Value ^†^
Household Asset Variables						
Medium Asset	−0.085	0.110	−0.001	0.961	−0.016	0.305
High Asset	0.001	0.988	−0.002	0.904	0.040 *	0.019
Media Exposure Variables						
Newspaper	0.022 *	0.033	0.006	0.316	0.012 *	0.024
Radio	0.005	0.727	−0.000188	0.979	0.001	0.874
Television	0.031	0.098	0.028 ***	0.000	0.032 ***	0.000
* **Dependency Variables** *						
High Asset * # above 60	−0.013	0.499	−0.002	0.845	−0.002	0.833
# above 60	0.030 *	0.032	0.013 **	0.006	0.015 **	0.002
# between 0 and 5	−0.027 ***	0.000	−0.005	0.075	−0.012 ***	0.000
# between 6 and 15	−0.001	0.899	0.005 *	0.018	0.002	0.206
*Caste*						
SC	−0.032	0.568	0.055 ***	0.000	0.050 **	0.001
ST	−0.028	0.623	0.034 *	0.018	0.034 *	0.027
OBC	−0.006	0.918	0.011	0.416	0.012	0.401
SC * Medium Asset	0.064	0.335	−0.010	0.579	−0.008	0.679
SC * High Asset	0.0004251	0.995	−0.011	0.613	−0.056 **	0.006
ST * Medium Asset	0.064	0.421	0.008	0.734	0.006	0.808
ST * High Asset	0.008	0.906	−0.031	0.241	−0.054 *	0.019
OBC * Medium Asset	0.062	0.351	0.004	0.802	0.015	0.389
OBC * High Asset	−0.044	0.460	0.013	0.503	−0.049 **	0.003
* **Control Variables** *						
Hindu	−0.019	0.400	0.035 *	0.010	0.015	0.210
Muslim	−0.053 *	0.036	0.022	0.220	−0.006	0.675
Female-headed HH	−0.010	0.470	−0.005	0.472	−0.007	0.294
HH head’s age	0.006 *	0.024	0.013 ***	0.000	0.011 ***	0.000
Age square	−0.000047	0.079	−0.0001142 ***	0.000	−0.000092 ***	0.000
HH head’s education	0.003 *	0.031	−0.00026	0.673	0.001	0.144
Agriculture: Male	0.014	0.529	−0.002	0.854	0.003	0.756
Non-Agriculture: Male	0.028	0.130	0.005	0.603	0.011	0.181
Agriculture: Female	0.016	0.477	0.027 ***	0.000	0.030 ***	0.000
Non-Agriculture: Female	0.024 *	0.023	0.028 ***	0.000	0.028 ***	0.000
Region	N.A.	N.A.	N.A.	N.A.	−0.041 ***	0.000
No. of observations	21850		51769		73619	
Pseudo R square	0.1451		0.2448		0.2019	
Log pseudolikelihood	−13.6521		−22.4992		−36.5230	

Note: ^†^ Sampling weights are used. Standard Errors are robust to heteroscedasticity at the cluster level. The marginal effect is for the discrete change of dummy variable from 0 to 1. Base Category for State Fixed Effects = Karnataka. Union Territories Chandigarh, Dadra and Nagar Haveli, Daman and Diu, Lakshadweep and Puducherry clubbed together. Delhi excluded from analysis in Rural sample. * in variable names indicates interaction, and # is used as a symbol for the word “number”. * *p* < 0.05, ** *p* < 0.01, *** *p* < 0.001.

**Table 5 healthcare-11-01343-t005:** MNL model estimates for health insurance choices in 2015–2016.

A. Outcome 1: Public H.I.						
	Urban		Rural		Combined	
	RRR	*p*-Value ^†^	RRR	*p*-Value ^†^	RRR	*p*-Value ^†^
Household Asset Variables						
Medium Asset	0.577	0.094	0.998	0.985	0.921	0.446
High Asset	0.610	0.117	0.841	0.169	0.886	0.283
Media Exposure Variables						
Newspaper	1.042	0.503	1.002	0.969	1.014	0.673
Radio	0.988	0.884	0.967	0.500	0.974	0.562
Television	1.182	0.175	1.214 ***	0.000	1.241 ***	0.000
Dependency Variables						
Highest Asset * # above 60	0.866	0.254	0.987	0.883	0.935	0.359
# above 60	1.121	0.213	1.085 *	0.021	1.082 *	0.025
# between 0 and 5	0.844 ***	0.000	0.971	0.129	0.932 ***	0.000
# between 6 and 15	1.013	0.643	1.039 **	0.005	1.029 *	0.022
Caste						
SC	0.752	0.376	1.416 **	0.001	1.319 **	0.006
ST	0.790	0.481	1.250 *	0.027	1.185	0.087
OBC	0.824	0.567	1.062	0.524	1.038	0.696
SC * Medium Asset	1.496	0.261	0.962	0.772	0.972	0.824
SC * High Asset	1.493	0.257	1.008	0.960	0.930	0.623
ST * Medium Asset	1.519	0.334	1.043	0.812	1.065	0.691
ST * High Asset	1.818	0.111	0.826	0.342	1.062	0.724
OBC * Medium Asset	1.521	0.249	1.027	0.828	1.102	0.405
OBC * High Asset	1.261	0.504	1.156	0.279	1.013	0.917
* **Control Variables** *						
Hindu	0.918	0.501	1.323 **	0.004	1.140	0.088
Muslim	0.865	0.335	1.197	0.154	1.070	0.486
Female-headed HH	0.887	0.175	0.965	0.467	0.932	0.113
HH head’s age	1.026	0.207	1.099 ***	0.000	1.073 ***	0.000
Age square	1.000	0.369	0.999 ***	0.000	0.999 ***	0.000
HH head’s education	0.992	0.284	0.993	0.091	0.992 *	0.036
Agriculture: Male	0.967	0.838	0.992	0.906	0.994	0.918
Non-Agriculture: Male	1.066	0.595	1.038	0.571	1.047	0.425
Agriculture: Female	1.085	0.532	1.203 ***	0.000	1.197 ***	0.000
Non-Agriculture: Female	1.122	0.075	1.189 ***	0.000	1.168 ***	0.000
Region	N.A.	N.A.	N.A.	N.A.	0.715 ***	0.000
**B. Outcome 2: CBHI**						
	**Urban**		**Rural**		**Combined**	
	** *RRR* **	** *p-* ** **Value ^†^**	**RRR**	** *p-* ** **Value ^†^**	**RRR**	** *p-* ** **Value ^†^**
Household Asset Variables						
Medium Asset	414,263.200 ***	0.000	0.237	0.187	0.322	0.209
High Asset	700,030.700 ***	0.000	0.478	0.289	1.027	0.969
Media Exposure Variables						
Newspaper	1.773	0.060	1.647	0.211	1.834	0.075
Radio	0.930	0.782	0.092 ***	0.000	0.602	0.215
Television	0.544	0.256	0.823	0.664	0.545	0.132
Dependency Variables						
Highest Asset * # above 60	1.884	0.161	1.987	0.302	0.897	0.850
# above 60	1.336	0.295	0.913	0.809	0.906	0.754
# between 0 and 5	0.838	0.239	1.147	0.340	1.229	0.135
# between 6 and 15	1.039	0.656	0.901	0.506	0.958	0.729
Caste						
SC	263,280.000 ***	0.000	0.409	0.234	0.418	0.241
ST	0.605	0.154	0.116	0.076	0.117	0.069
OBC	289,420.400 ***	0.000	0.336	0.133	0.494	0.339
SC * Medium Asset	0.000003 ***	0.000	1.851	0.704	6.389	0.169
SC * High Asset	0.000001 ***	0.000	3.993	0.218	1.292	0.796
ST * Medium Asset	0.912	0.933	73.535 *	0.017	39.110 *	0.033
ST * High Asset	1.552	0.549	0.001 ***	0.000	11.839	0.069
OBC * Medium Asset	0.000004 ***	0.000	12.446 *	0.048	7.178	0.062
OBC * High Asset	0.000002 ***	0.000	2.810	0.297	1.788	0.493
Control Variables						
Hindu	0.486	0.133	0.862	0.769	0.868	0.819
Muslim	0.173 **	0.004	0.485	0.378	0.333	0.191
Female-headed HH	1.539	0.202	1.491	0.487	1.135	0.781
HH head’s age	1.222 ***	0.000	1.164	0.184	1.296 *	0.016
Age square	0.998 ***	0.000	0.998	0.190	0.998 *	0.025
HH head’s education	0.995	0.877	1.009	0.833	1.036	0.324
Agriculture: Male	1.618	0.126	2.500	0.064	1.563	0.274
Non-Agriculture: Male	1.716	0.069	1.201	0.712	0.775	0.532
Agriculture: Female	1.331	0.566	0.564	0.248	0.583	0.218
Non-Agriculture: Female	1.329	0.326	0.290	0.098	1.089	0.824
Region	N.A.	N.A.	N.A.	N.A.	1.331	0.426
**C. Outcome 3: Private H.I.**						
	**Urban**		**Rural**		**Combined**	
	**RRR**	** *p-* ** **Value ^†^**	**RRR**	** *p-* ** **Value ^†^**	**RRR**	** *p-* ** **Value ^†^**
Household Asset Variables						
Medium Asset	68654.040 ***	0.000	4.246 **	0.005	3.509 *	0.011
High Asset	616760.200 ***	0.000	11.242 ***	0.000	18.188 ***	0.000
Media Exposure Variables						
Newspaper	1.477 *	0.025	1.322	0.082	1.433 **	0.003
Radio	1.229	0.189	1.166	0.411	1.172	0.209
Television	3.399 *	0.021	0.909	0.726	1.476	0.089
Dependency Variables						
Highest Asset * # above 60	0.923	0.745	1.236	0.427	1.051	0.775
# above 60	1.249	0.210	0.902	0.545	1.074	0.558
# between 0 and 5	0.909	0.447	0.933	0.396	0.917	0.319
# between 6 and 15	0.946	0.409	0.948	0.440	0.943	0.246
* **Caste** *						
SC	86,983.790 ***	0.000	1.945	0.328	2.624	0.120
ST	0.957	0.880	1.197	0.769	1.776	0.347
OBC	93,107.950 ***	0.000	2.712 *	0.046	3.609 **	0.008
SC * Medium Asset	0.000018 ***	0.000	0.350	0.180	0.371	0.156
SC * High Asset	0.000007 ***	0.000	0.341	0.179	0.221 *	0.020
ST * Medium Asset	2.777	0.115	1.247	0.723	1.118	0.857
ST * High Asset	0.215 **	0.001	0.572	0.463	0.154 **	0.006
OBC * Medium Asset	0.00002 ***	0.000	0.356	0.080	0.372	0.073
OBC * High Asset	0.000005 ***	0.000	0.390	0.087	0.148 ***	0.000
Control Variables						
Hindu	1.275	0.396	1.392	0.352	1.439	0.121
Muslim	0.633	0.231	0.793	0.586	0.729	0.303
Female-headedHH	0.882	0.527	0.742	0.158	0.850	0.285
HH head’s age	0.997	0.909	0.998	0.946	1.001	0.951
Age square	1.000	0.611	1.000	0.659	1.000	0.598
HH head’s education	1.144 ***	0.000	1.072 ***	0.000	1.125 ***	0.000
Agriculture: Male	1.922	0.107	0.891	0.654	1.296	0.293
Non-Agriculture: Male	2.271 *	0.023	1.327	0.259	1.755 *	0.011
Agriculture: Female	1.082	0.892	0.994	0.979	1.012	0.959
Non-Agriculture: Female	1.067	0.636	1.417	0.060	1.128	0.291
Region	N.A.	N.A.	N.A.	N.A.	1.349 *	0.014
**D. Outcome 4: Others**						
	**Urban**		**Rural**		**Combined**	
	**RRR**	** *p-* ** **Value ^†^**	**RRR**	** *p-* ** **Value ^†^**	**RRR**	** *p-* ** **Value ^†^**
Household Asset Variables						
Medium Asset	N.A.	N.A.	0.868	0.704	0.994	0.986
High Asset	N.A.	N.A.	0.620	0.316	1.185	0.653
Media Exposure Variables						
Newspaper	N.A.	N.A.	1.333 *	0.037	1.458 **	0.005
Radio	N.A.	N.A.	1.057	0.759	0.954	0.776
Television	N.A.	N.A.	1.383	0.137	1.251	0.320
Dependency Variables						
Highest Asset * # above 60	N.A.	N.A.	1.023	0.943	1.575	0.098
# above 60	N.A.	N.A.	1.340 *	0.010	1.375 **	0.002
# between 0 and 5	N.A.	N.A.	1.009	0.903	0.934	0.324
# between 6 and 15	N.A.	N.A.	1.006	0.895	1.006	0.889
Caste						
SC	N.A.	N.A.	1.653	0.081	1.589	0.120
ST	N.A.	N.A.	1.014	0.965	1.016	0.962
OBC	N.A.	N.A.	1.211	0.531	1.248	0.480
SC * Medium Asset	N.A.	N.A.	0.418	0.062	0.379 *	0.032
SC * High Asset	N.A.	N.A.	1.525	0.420	0.325 *	0.010
ST * Medium Asset	N.A.	N.A.	0.675	0.472	0.651	0.417
ST * High Asset	N.A.	N.A.	1.207	0.750	0.884	0.859
OBC * Medium Asset	N.A.	N.A.	0.864	0.732	0.862	0.722
OBC * High Asset	N.A.	N.A.	1.109	0.833	0.528	0.108
Control Variables						
Hindu	N.A.	N.A.	0.665	0.138	0.545 *	0.025
Muslim	N.A.	N.A.	0.691	0.404	0.331 **	0.004
Female-headed HH	N.A.	N.A.	1.335	0.158	1.499	0.056
HH head’s age	N.A.	N.A.	1.068	0.088	1.105 **	0.002
Age square	N.A.	N.A.	0.999	0.119	0.999 **	0.003
HH head’s education	N.A.	N.A.	1.043 *	0.025	1.024	0.217
Agriculture: Male	N.A.	N.A.	0.945	0.806	1.067	0.725
Non-Agriculture: Male	N.A.	N.A.	1.157	0.487	1.306	0.103
Agriculture: Female	N.A.	N.A.	1.057	0.723	1.103	0.530
Non-Agriculture: Female	N.A.	N.A.	1.007	0.972	1.124	0.555
Region	N.A.	N.A.	N.A.	N.A.	0.746	0.090
No. of observations	21,592		51,506		72,660	
Pseudo R square	0.183		0.251		0.224	
Log pseudolikelihood	−162.217		−247.842		−415.239	

Note: ^†^ Sampling weights are used. Standard Errors are robust to heteroscedasticity at the cluster level. Base Category for State Fixed Effects = Karnataka. CBHI and Other H.I. are clubbed for urban sample. Union Territories Chandigarh, Dadra and Nagar Haveli, Daman and Diu, Lakshadweep and Puducherry clubbed together. Delhi excluded from analysis in Rural sample. * in variable names indicates interaction, and # is used as a symbol for the word “number”. * *p* < 0.05, ** *p* < 0.01, *** *p* < 0.001.

## Data Availability

The data presented in this study are available upon request from the Demographic and Health Surveys (DHS) website.

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
