# Peer review of "Wealth Status and Health Insurance Enrollment in India: An Empirical Analysis"

_healthcare, 2023, doi:10.3390/healthcare11091343_

Round 1

Reviewer 1 Report

The paper analyses the outcomes from a survey about health insurance in India, with the purpose to assess the effect of households’ socio-demographic on health insurance enrollment in this country. The paper extends a similar investigation by Chakrabarti and Shankar (2105), considering updated data. The interest of such an extension relies in particular on the fact that further reforms have been implemented.

Although not contributing any methodological novelty on the subject, the paper is interesting, well written and with an accurate discussion.

However, the Authors were naïve in some respects.

-       The Abstract needs to be rewritten, following a more fluid narrative (for instance, avoiding to arrange it into Sections).

-       It is uncomfortable to have so many footnotes, and to have them listed at the end of the paper (this latter matter is perhaps a matter of Journal’s template, however).

-       Some instructions from the template have not been removed: lines 207-213, 372-375.

-       Tables 1 and 2 are too extended to be placed in the main text. The Authors should consider reporting in the main text only a summary, moving the detailed information to an Appendix.

-       Section 4 is a little bit repetitive in respect of previous comments, in particular at the beginning.

-       Typos: The reference of Foonote 1 is not in the superscript style (line 38); blank spaces are missing in many places (for example, see lines 193, 197); line 223 should be aligned on the left-hand side.

Author Response

Thank you for your valuable feedback. Our point by point response is attached.

Reviewer 2 Report

Thank you for the opportunity to review the manuscript. It is a well-written manuscript, and I enjoyed reading it. However, I have a few suggestions to be addressed before final acceptance:

Some typos (i.e., page 1 -line 38) and some expressions could be clearer in the manuscript. Please correct it.

In the material and method section, figures 1 and 2 and its description do not fit into the "materials ad method section", so they should be moved to the result or explained in the discussion section.

The discussion section should be strengthened in explaining the current "health insurance status" in India and how the findings could help support the policy development for ensuring health equity in India. Please also mention the limitations and practical guidelines for strengthening health insurance coverage in India in a more detailed way so that policymakers can take it as a reference to address this problem.

Please also elaborate in the discussion section on the reason for the increased health insurgence coverage along with increasing social inequalities in India.

As India is a big country with a larger population, it may not be beneficial to use all Indian data and give a conclusion. Rather than this, if authors compare the state-wise health insurance enrollment and inequalities and they provide results, it would be more helpful for policymakers to comprehend the status and patterns. If so, then, the study results could be more beneficial for health policymakers and have a higher impact. Therefore, if the authors reanalyze the data from each state and provide results, it would be more comprehensive and helpful for health authorities and policymakers to take appropriate action further.

The study also does not cover recent studies on "inequalities" in health in South Asia, particularly India.  Most studies referred to in the study are old and outdated. Please also update on the most recent studies conducted on health inequalities in South Asia, particularly in India.

From a reviewer's perspective, I am curious how missing values were excluded from the analysis; it also needs to be clarified. For example, the authors mentioned, "For analysis purposes, we ignore 'do not know' and 154 missing responses." Such a practice may somewhat provide "biased" results/findings. Further, how missing values in independent variables were handled/treated is more important for clarification. It is missing in the manuscript. Please explain and provide an appropriate rationale.

A separate conclusion section is required.

Author Response

Thank you for your valuable feedback. Our point-by-point response is attached.

Reviewer 3 Report

Dear Authors,

Congratulations on a well-written paper. The introduction describes the problem in a comprehensive and understandable way. The methodology of the study is also coherent and well thought out. 

I would only suggest adding a section on the direction of future research in the discussion. The Authors mention this at the beginning of the discussion ("Future research directions may also be highlighted."), but I do not find a corresponding passage in the rest of the text. 

Furthermore, your manuscript lacks conclusions.

Good luck

Author Response

(The authors gave the same response as above.)

Reviewer 4 Report

The purpose of this paper is to investigate the factors that are related to health insurance coverage status of households in India. The study utilizes a cross-sectional data set from a household survey. Authors employ probit and logistic regression models for empirical analysis. Results indicate that wealth level, member composition, caste type, and head characteristics of households are associated with health insurance coverage of households. There are also regional variations in levels and determinants of health insurance coverage.

I have read the paper with interest. Although the study reveals some insights and empirical evidence on the subject matter, I have concerns on the manuscript, which are listed below. 

Comments:

·         The manuscript does not provide theoretical/conceptual background for most of the predictor variables.

·         Generally, the literature provides analysis of health insurance choices at individual level. The manuscript does not provide a discussion on reasons for its analysis at household level. What are the specific conditions for the case of India that drive insurance coverage at household level rather than individual level?

·         A methodological concern for the study is the nested nature of the data. The data are obtained at household-level. There are also significant state-level variations in the data. The empirical estimation framework of the manuscript does not account for the hierarchical structure.

·         There are potential omitted variable biases. Are any specific regional control variables at state level (such as level of health services/expenditures) that should be included in the models?

·         The paper does not provide a table for descriptive statistics and distribution tests for the dependent variables.

·         A potentially serious empirical limitation of the manuscript lies in the modelling choices. Are choices of probit and logistic models in line with skewness and distribution of the variables of interest?

·         Authors state that there are missing data and observations. Is there any systematic bias for them?

·         The first paragraph of section 2.3 is not related to manuscript content (Lines:207-213).

Author Response

(The authors gave the same response as above.)

Reviewer 5 Report

Comments to Author - healthcare-2276551
The manuscript “Wealth status and health insurance enrollment in India: An empirical analysis of National
Family Health Survey-4 (2015-16) analyzed National Family Health Survey data to assess the health
insurance coverage in India in the last fifteen years.

The problem considered is interesting and important and the findings are likely to be interesting to others
working in this field. I encourage the authors to consider the following comments to strengthen the
manuscript (in the order of importance):

1. Since this study replicated Chakrabarti and Shankars previous study using the latest date, the
authors should include a summary table to show the comparison of the results from these two
studies.

2. Furthermore, the authors may want to consider a similar approach (summary table) for the
comparison of the two modeling approaches they utilized in this study, Probit and MNL.

3. Line 207-213 or Line 372-375 doesn’t seem to belong to this manuscript.

4. Table 2 should also include the significance level of the results, as in Table 1.

Considering the above-mentioned comments, I suggest a minor revision.

Author Response

(The authors gave the same response as above.)

Round 2

Reviewer 4 Report

The manuscript is significantly revised. Although it has some limitations, the current version makes acceptable for publication.

Author Response

Thank you for your valuable feedback. It has undoubtedly helped us to improve the manuscript.